# Metabolic Regulation and Lipidomic Remodeling in Relation to Spermidine-induced Stress Tolerance to High Temperature in Plants

**DOI:** 10.3390/ijms232012247

**Published:** 2022-10-13

**Authors:** Zhou Li, Bizhen Cheng, Yue Zhao, Lin Luo, Yan Zhang, Guangyan Feng, Liebao Han, Yan Peng, Xinquan Zhang

**Affiliations:** 1College of Grassland Science and Technology, Sichuan Agricultural University, Chengdu 611130, China; 2Institute of Turfgrass Science, Beijing Forestry University, Beijing 100083, China

**Keywords:** signaling pathway, heat stress, metabolomics, phospholipids, glycoglycerolipids, sphingolipids, unsaturation index

## Abstract

Beneficial effects of spermidine (Spd) on alleviating abiotic stress damage have been explored in plants for hundreds of years, but limited information is available about its roles in regulating lipids signaling and metabolism during heat stress. White clover (*Trifolium repens*) plants were pretreated with 70 μM Spd and then subjected to high temperature (38/33 °C) stress for 20 days. To further investigate the effect of Spd on heat tolerance, transgenic *Arabidopsis*
*thaliana* overexpressing a *TrSAMS* encoding a key enzyme involved in Spd biosynthesis was exposed to high temperature (38/33 °C) stress for 10 days. A significant increase in endogenous Spd content in white clover by exogenous application of Spd or the *TrSAMS* overexpression in *Arabidopsis*
*thaliana* could effectively mitigate heat-induced growth retardation, oxidative damage to lipids, and declines in photochemical efficiency and cell membrane stability. Based on the analysis of metabolomics, the amino acids and vitamins metabolism, biosynthesis of secondary metabolites, and lipids metabolism were main metabolic pathways regulated by the Spd in cool-season white clover under heat stress. Further analysis of lipidomics found the *TrSAMS*-transgenic plants maintained relatively higher accumulations of total lipids, eight phospholipids (PC, phosphatidylcholine; PG, phosphatidylglycerol; PS, phosphatidylserine; CL, cardiolipin; LPA, lysophosphatidic acid; LPC, lyso phosphatidylcholine; LPG, lyso phosphatidylglycerol; and LPI, lyso phosphatidylinositol), one glycoglycerolipid (DGDG, digalactosyl diacylglycerol), and four sphingolipids (Cer, ceramide; CerG2GNAc1, dihexosyl N-acetylhexosyl ceramide; Hex1Cer, hexosyl ceramide; and ST, sulfatide), higher ratio of DGDG: monogalactosyl diacylglycerol (MGDG), and lower unsaturation level than wild-type *Arabidopsis*
*thaliana* in response to heat stress. Spd-induced lipids accumulation and remodeling could contribute to better maintenance of membrane stability, integrity, and functionality when plants underwent a long period of heat stress. In addition, the Spd significantly up-regulated PIP2 and PA signaling pathways, which was beneficial to signal perception and transduction for stress defense. Current findings provide a novel insight into the function of Spd against heat stress through regulating lipids signaling and reprograming in plants.

## 1. Introduction

For most of plant species adapting to temperate climate, high environmental temperature disturbs metabolic homeostasis, resulting in the retardation of seed germination, plant growth, development, and reproduction [1]. Heat-induced crop failure and other agricultural losses will further increase due to the intensification of global warming [2]. Plants have developed complex defensive strategies to alleviate heat damage including alterations in endogenous hormones or plant growth regulators (PGRs) [3]. Spermidine (Spd) is one of the commonest polyamines acting as a critical PGR for regulation of seed germination, plant growth, and stress tolerance [4]. It has been reported that enhanced Spd biosynthesis and metabolism are positive responses to various abiotic stresses such as heat, drought, and salt stress, since Spd plays a key role in regulating cell signal transduction, osmotic adjustment, antioxidant metabolism, and membrane protection in plants [5,6,7,8]. Significant increase in endogenous Spd content by exogenous application of Spd or transgenic approach could effectively alleviate heat damage owing to enhanced antioxidant capacity in rice (*Oryza sativa*) leaves [9], starch metabolism in rice seeds [10], photosynthetic performance in tomato (*Lycopersicon esculentum*) leaves [11], and transcriptional regulation in ripening tomato fruit [12]. Our previous studies also demonstrated that foliar application of the polyamine such as Spd or spermine (Spm) mitigated adverse effects of heat stress in white clover (*Trifolium repens*) or creeping bentgrass (*Agrostis stolonifera*) associated with changes in antioxidant defense or heat shock pathways, respectively [13,14].

Oxidative burst and overaccumulation of oxidizing agents damage the cell membrane system, which is one of the major stress symptoms in plants subjected to a prolonged period of heat stress and other abiotic stresses [15,16,17,18]. Therefore, the maintenance of the structural integrity of the membrane is beneficial to the stress defense system [19]. Lipids such as phospholipids (Phls), glycoglycerolipids (Glls), and sphingolipids (Spls) are not only important components of the membrane system, but also exhibit a variety of roles in cell signaling, protein modification, energy storage, and membrane anchoring in plants [20]. The fact that there are more than 200,000 lipid molecular species in biological systems increases the difficulty of lipid research over a long period of time [21]. As a new approach, lipidomics provides a rapid and sensitive strategy to reveal vital function of diverse lipid species in living beings [22]. A recent study demonstrated that a significantly higher polyunsaturated linolenic acid (18:3) content and a lower lipid unsaturation level were in favor of heat adaptation based on comparative lipidomic analysis between heat-tolerant soybean (*Glycine max*) genotype DS25-1 and heat-susceptible genotype DT97-4290 [23]. General analysis of the lipid profile revealed that enhanced triacylglycerol accumulation and decline in membrane polyunsaturated fatty acids were responsible for better adaptation of *Arabidopsis thaliana* to heat stress [24]. In addition, Spd in association with changes in lipids has also been reported in plant species. For example, the Spd induced significant increases in phosphatidylinositol 4-phosphate (PIP) and phosphatidylinositol 4,5-bisphosphate (PIP2) in *Brassica oleracea* seedlings under normal condition [25]. Quick accumulation of stearic acid, linoleic acid, and linolenic acid could be significantly inhibited by exogenous Spd in leaves of cucumber (*Cucumis sativus*) under salt stress [26]. However, potential roles of Spd in regulating lipid signaling and global lipid compositions in response to environmental stress, especially high temperatures, are not well elucidated in plant species.

Objectives of this study were to investigate effects of exogenous application of Spd on physiological changes and metabolites profile in white clover under heat stress and to further reveal the Spd-regulated lipids signal and remodeling in transgenic *Arabidopsis thaliana* overexpressing a *TrSAMS* encoding a *S*-adenosylmethionine synthetase from white clover involved in Spd biosynthesis in response to heat stress. Current findings facilitated a comprehensive understanding of an underlying mechanism of Spd-mediated stress adaptation associated with the alleviation of growth retardation, enhanced phosphatidylinositol (PI) and phosphatidic acid (PA) signaling pathways, and maintenance of membrane integrity and functionality in plants exposure to high temperature environment.

## 2. Results

### 2.1. Stress Tolerance and Metabolomics Regulated by Spermidine in White Clover under Heat Stress

Heat stress significantly inhibited white clover growth, but this adverse effect could be alleviated by the application of Spd under heat stress (Figure 1A). As compared with the control, endogenous Spd content significantly increased under normal and heat stress conditions due to the Spd application (Figure 1B). Shoot length (SL) of Spd-pretreated or non-pretreated plants was significantly inhibited by heat stress; however, Spd-pretreated plants exhibited significantly higher SL than non-pretreated plants under heat stress (Figure 1C). There were no significant differences in root length (RL) among three treatments (C, C + Spd, and H), and the Spd significantly increased the RL under heat stress (Figure 1D). In spite of the Spd application, photochemical efficiency (Fv/Fm) and performance index on absorption basis (PIABS) significantly declined in both of Spd- and non-pretreated plants after 20 days of heat stress, but this declined trend could be significantly alleviated by the exogenous Spd application (Figure 1E,F). Heat-induced increases in malondialdehyde (MDA) content and electrolyte leakage (EL) could also be alleviated significantly by the Spd (Figure 1G,H). 

A large number of differentially accumulated metabolites (DAMs) were identified in leaves of white clover in response to the Spd application and heat stress (Appendix A). Except for global and overview maps, most of DAMs were involved in biosynthesis of other secondary metabolites, amino acid metabolism, and metabolism of cofactors and vitamins based on the analysis of KEGG function annotation (Appendix A). Lipid maps of DAMs showed that most of the identified DAMs were associated with flavonoids, steroids, isoprenoids, and fatty acids and conjugates (Appendix A). Volcano plot of DAMs indicated the distribution of DAMs in each group, as the abscissa presented log_2_ fold change (FC) of DAMs and the ordinate presented significant level (-log_10_
*p*-value) (Appendix A). A total of 24, 27, 217, and 203 DAMs were identified in the CS vs. C, HS vs. H, H vs. C, ands HS vs. C, respectively (Figure 2A). Propanoic acid, butanoic acid, triacylglycerol (TAG) (18:0-18:1-22:3), valine, serine, pyroglutamic acid, nicotinamide, pyridoxal, abscisic acid, hydroxyecdysome, methyl dihydrojasmonate, and crocetin significantly increased, whereas docosahexaenoic acid, TAG (18:1-18:1-20:1), and oxidized glutathione significantly decreased in the HS vs. H (Figure 2B).

### 2.2. Endogenous Spermidine Level and Stress Tolerance Regulated by TrSAMS Overexpression in Arabidopsis thaliana under Heat Conditions

Transgenic *Arabidopsis* overexpressing the *TrSAMS* (OE4 and OE5) grew bigger than the wild type (WT) under normal condition and also showed better growth under heat stress (Figure 3A). Endogenous Spd contents were significantly higher in the OE4 and OE5 as compared to that in the WT under normal and heat conditions (Figure 3B). Although heat stress decreased fresh weight (FW) and dry weight (DW) of above-ground biomass, leaf length (LL), and leaf width (LW) of WT and transgenic plants, the OE4 and OE5 exhibited significantly higher FW and DW of above-ground biomass, LL, and LW than the WT under normal and heat conditions (Figure 3C,D,F,G). However, there was no significant difference in DW to FW ratio among WT, OE4, and OE5 (Figure 3E). Transgenic plants maintained significantly higher total chlorophyll (Chl), Chl a, Chl b, and Chl a/b ratio than the WT under heat stress (Figure 4A–D). Fv/Fm and PIABS were not significant differences between the WT and transgenic plants under normal condition and declined significantly under heat stress (Figure 4E,F). Transgenic plants exhibited a 43% or 120% increase in Fv/Fm or PIABS compared to the WT under heat stress, respectively (Figure 4E and 4F). Under normal conditions, the *TrSAMS* overexpression did significantly affect MDA content and EL in leaves. MDA content and EL rose steeply in all plants due to heat damage, but transgenic plants showed 27% and 32% decrease in MDA content or EL than the WT under heat stress, respectively (Figure 4G and 4H).

### 2.3. Heat-Map and Principal Component Analysis Based on 31 Lipid Classes in Arabidopsis thaliana in Response to Spermidine and Heat Stress

Heat-map of different lipid classes showed that most of lipid classes were down-regulated in WT and OE4 in response to 3 h or 10 d of heat stress, but PI, PIP2, lysophosphatidic acid (LPA), lyso phosphatidylinositol (LPI), and ceramide (Cer) were significantly up-regulated by heat stress in the OE4 (Figure 5A). First and second principle component explained 41.0% and 29.7% variation among 31 lipid classes in response to the Spd and heat stress in leaves of *Arabidopsis thaliana*, respectively (Figure 5B). The principal component analysis (PCA) found that the PIP2, ganglioside-disialo trihexosyl ceramide (GD2), Cer, dihexosyl N-acetylhexosyl ceramide (CerG2GNAc1), phosphatidylglycerol (PG), phosphatidylserine (PS), lyso phosphatidylcholine (LPC), lyso phosphatidylethanolamine (LPE), digalactosyl monoacylglycerol (DGMG), and digalactosyl diacylglycerol (DGDG) were the main lipid classes affected by the Spd and heat stress (Figure 5B).

### 2.4. Changes in Total Lipids, Phospholipids, Glycoglycerolipids, and Sphingolipids Regulated by Spermidine in Arabidopsis thaliana under Heat Stress

Heat stress index (HSI) of total lipids significantly declined in WT and OE4 from 3 h to 10 d of heat stress, but OE4 exhibited significantly higher HSI of total lipids than WT at 3 h and on the 10th d of heat stress (Figure 6A). OE4 exhibited 36% and 12% increase in HSI of total Phl compared to WT at 3 h and on 10th d of heat stress, respectively (Figure 6A). There was no significant difference in the HSI of total Gll between WT and OE4 at 0 h or on 10th d of heat stress (Figure 6A). OE4 had significantly higher HSI of total Spl than WT at 3 h of heat stress, but no significant difference was detected on 10th d of heat stress (Figure 6A). 

HSI of PA and phosphatidylcholine (PC) were not significantly different between WT and OE4 at 3 h of heat stress, whereas OE4 had significantly higher HSI of PA and lower HIS of PC than WT on the 10th d of heat stress (Figure 6B). OE4 maintained significantly higher or lower HIS of phosphatidylethanolamine (PE) than WT at 3 h and on 10th d of heat stress, respectively (Figure 6B). Although no significant difference in HSI of PG or LPE was detected between OE4 and WT on the 10th d of heat stress, OE4 showed significantly higher HSI of PG and LPE than WT at 3 h of heat stress. HSI of PS, PI, PIP2, and LPC were maintained at a significantly higher level in OE4 as compared to WT during heat stress. OE4 also exhibited significantly higher HSI of PIP, cardiolipin (CL), LPG or LPI at 3 h of heat stress as well as higher HSI of LPA than WT on 10th d of heat stress. There was no significant difference in HSI of phosphatidylinositol triphosphate (PIP3) or mono-lyso cardiolipin (MLCL) between OE4 and WT during heat stress (Figure 6B). 

HSI of DGDG, monogalactosyl diacylglycerol (MGDG), and GD2 was not significantly different between WT and OE4 at 3 h or on 10th d of heat stress (Figure 6C). Significantly lower HSI of DGMG and monogalactosyl monoacylglycerol (MGMG) at 3 h of heat stress and significantly lower HSI of MGMG, ceramide phosphate (CerP), dihexosyl ceramide (Hex2Cer), trihexosyl ceramide (Hex3Cer), ganglioside-disialo tetrahexosyl ceramide (GD1a), and ganglioside-disialo dihexosyl ceramide (GD3) on the 10th d of heat stress were detected in WT as compared to that in OE4 (Figure 6C). A 45%, 11%, and 49% increase in the HSI of Cer, hexosyl ceramide (Hex1Cer), and ganglioside-trisialo dihexosyl ceramide (GT3) was detected in OE4 as compared to WT at 3 h of heat stress, respectively (Figure 6C). OE4 also showed significantly higher HSI of CerG2GNAc1 or sulfatide (ST) than WT at 3 h and on 10th d of heat stress (Figure 6C).

### 2.5. Lipid Molecular Species, Ratios of PC:PE and DGDG:MGDG, and the Unsaturation Index of Lipids Regulated by Spermidine in Arabidopsis thaliana under Heat Stress

Figure 7 shows the changes in lipid molecular species of different lipid classes (PA, PC, PE, PG, PI, CL, LPA, LPC, LPE, LPG, LPI, MLCL, DGDG, MGDG, DGMG, MGMG, Cer, CerP, CerG2GNAc1, GT3, Hex1Cer, Hex2Cer, Hex3Cer, and GD3) when the transgenic plants were compared with the WT (OE4 vs. WT) at 0 h, 3 h, or 10 d of heat stress. Ratio of PC:PE of WT or OE4 did not significantly change during heat stress (Figure 8A). OE4 exhibited a significantly higher ratio of DGDG:MGDG than WT on the 10th d of heat stress (Figure 8B).

The unsaturation index of total lipids of WT significantly increased at 3 h of heat stress and maintained a significant level after 10 d of heat stress (Figure 8C). However, the unsaturation index of OE4 did alter during heat stress (Figure 8C). Unsaturation index of total Phl of WT and OE4 also did not change significantly when subjected to heat stress for 10 d, but OE4 showed a significantly lower unsaturation index of total Phl than WT at 3 h of heat stress (Figure 8D). Heat stress (3 h and 10 d) induced significant increase in the unsaturation index of total Gll in WT, whereas it did not affect OE4 (Figure 8E). Significantly lower unsaturation index of total Gll was observed in OE4 as compared to that in WT during heat stress (Figure 8E). Unsaturation index of total Spl did not change in WT at 3 h of heat stress, but significantly decreased after 10 d of heat stress (Figure 8F). Heat stress induced a gradual decline in unsaturation index of total Spl in OE4 during heat stress (Figure 8F). OE4 exhibited significantly lower unsaturation index of total Spl than WT at 3 h of heat stress (Figure 8F).

### 2.6. Key Genes Involved in PI and PA Metabolic Pathways Regulated by Spermidine in Arabidopsis thaliana under Heat Stress

Heat stress induced significant increases in transcript levels of *phosphatidylinositol 4-kinase alpha* (*AtPI4K-alpha*), *phosphatidylinositol phosphate 5-kinase 1* (*AtPIP5K1*), *phospholipase C 5* (*AtPLC5*), *diacylglycerol kinase 2* (*AtDGK2*), and *phospholipase D delta* (*AtPLD-delta*) in wild type and transgenic plants (Figure 9). Transgenic plants exhibited significantly higher expression levels of these genes than the wild type in response to heat stress (Figure 9). Figure 10 showed regulatory pathways induced by Spd associated with alterations in metabolic homeostasis, lipids signaling, and lipidomic reprogramming in favor of thermotolerance based on the summary of our current conclusions.

## 3. Discussion

High temperature stress is one of the major threats for most plants only adapting to temperate climate [27]. Increasing evidence has suggested that Spd exhibits a positive role in alleviating growth retardation, leaf senescence, and photoinhibition against heat stress in various plant species including white clover [9,10,14,28]. These previous studies were consistent with our current findings which showed that a significant increase in endogenous Spd content in white clover by exogenous application of Spd or in transgenic *Arabidopsis thaliana* overexpressing the *TrSAMS* enhanced the heat tolerance of white clover or *Arabidopsis thaliana* through maintaining root and shoot growth, photosynthesis, and cell membrane stability under high temperature condition. Metabolome analysis demonstrated that amino acids and vitamin metabolism, biosynthesis of secondary metabolites, and lipid metabolism were main metabolic pathways affected by the Spd in leaves of white clover suffering from heat stress. Accumulation and conversion of amino acids such as valine, serine, and pyroglutamic acid are adaptive responses to abiotic stress due to their roles in osmotic adjustment and biosynthesis of proteins and secondary metabolites [29,30]. Nicotinamide is a derivative of vitamin B3 that exhibits multiple functions of anti-aging, growth promotion, and stress defense in plants [31,32]. It has been found that foliar application of nicotinamide prevented biomass loss and lipid peroxidation in faba bean (*Vicia faba*) under salt stress [33]. As an essential component of vitamin B6, pyridoxal is involved in fatty acid metabolism, chlorophyll and auxin biosynthesis, and antioxidant under stressful condition [34,35]. In addition, regulatory roles of secondary metabolites such as abscisic acid (ABA), methyl dihydrojasmonate (MDJ), and hydroxyecdysone in stress tolerance have been reported widely in plants. For example, the ABA could mitigate heat-induced oxidative damage in *Arabidopsis thaliana* [36] and reed (*Phragmites communis*) [37]. Exogenous application of hydroxyecdysone protected rice against heat stress [38], and the MDJ could delay drought damage in tall fescue (*Festuca arundinacea*) [39]. Hence, Spd-induced alleviation of adverse effects in white clover could be related to increased accumulation of amino acids (valine, serine, and pyroglutamic acid), vitamins and their derivative (nicotinamide and pyridoxal), and secondary metabolites (ABA, MDJ, and hydroxyecdysone) under heat stress.

Previous studies have found that alterations in contents and composition of lipids, fatty acids, and TAG have a close association with heat tolerance in plant species [24,40,41]. In the current study, Spd-regulated DAMs included unsaturated fatty acid (propanoic acid and butanoic acid), saturated fatty acid (docosahexaenoic acid), and TAG (18:0-18:1-22:3 and 18:1-18:1-20:1) in leaves of white clover subjected to heat stress. To further elucidate potential association between Spd-regulated heat tolerance and lipids metabolism, lipidomics was conducted in wild or *TrSAMS*-transgenic *Arabidopsis thaliana* differing in endogenous Spd content. Results showed that *TrSAMS*-transgenic plants maintained relatively higher total lipids, total Phl, and total Spl contents as compared to the wild type under heat stress, which indicated that the change in endogenous Spd level indeed induced alteration in lipids metabolism. Heat stress triggers transition of membrane phase from a bilayer to non-bilayer resulting in hyper-fluidity of membranes due to a great deal of accumulation of unsaturated carbonyl groups in membranes [42]. Plants tend to decrease unsaturation levels of lipids to reduce lipid hyper-fluidity in favor of better maintenance of optimal membrane fluidity, integrity, and stability under high temperature, which is a common adaptation mechanism in plant species [43]. In addition, unsaturated lipids are easier to be oxidized than saturated ones under oxidative cellular environments such as high temperature condition [20]. A decline in lipid unsaturation level was advantageous to plants during a long-term heat stress, which has been proved in many plant species such as creeping bentgrass [44], *Arabidopsis thaliana* [45], rice [46], and soybean [23]. Unsaturation index of total lipids significantly increased in wild-type *Arabidopsis thaliana* but was kept on a normal level in *TrSAMS*-transgenic plants. OE4 also exhibited significantly lower unsaturation index of total Phl, Gll, and Spl than wild-type *Arabidopsis* in response to heat stress. These findings indicate that higher lipids saturation level regulated by Spd could be beneficial to maintain membrane stability and integrity in transgenic plants exposed to heat stress.

Phls are essential components of cellular membranes for forming lipid bilayers and also serve as secondary signaling molecules for stress signal transduction in plant cells exposed to high temperature stress [42]. Heat stress triggers PIP2 signaling pathway through activating key kinases including PI4K and PIP5K, as PI and PIP are two essential intermediates of the pathway. PA signaling generation mainly depends on two pathways: one is that structural Phls such as PC and PE are hydrolyzed by PLD to form the PA, and another is that the hydrolysates that are derived from hydrolytic reactions catalyzed by PLC are phosphorylated by DGK to produce the PA [47]. Both of PIP2 and PA are involved in regulations of membrane trafficking, organization cytoskeleton, gene expression, reactive oxygen species homeostasis, and hormonal response [48]. It has been found that thermotolerance could be enhanced significantly by the exogenous PA application associated with significant up-regulation of heat shock pathway in cool-season tall fescue [49]. Drought priming also improved heat tolerance of tall fescue by significantly increasing PI content throughout a period of heat stress [50]. Better adaption to heat stress could be achieved in heat-tolerant soybean genotype by up-regulating accumulation of PI [23]. In addition, a quick conversion of PI to PIP and then to PIP2 has been proved as one of the important adaptive responses to heat stress in plants [51]. Higher increases in PI, PIP, PIP2, and PA contents as well as transcript levels of genes (*AtPI4K-alpha*, *AtPIP5K1*, *AtPLC5*, *AtDGK2*, and *AtPLD-delta*) involved in PI and PA metabolism were detected in the OE4 as compared to wild-type *Arabidopsis thaliana* during heat stress in our current study, indicating Spd-regulated heat tolerance could be associated with the activation of PIP2 and PA signal pathways.

In addition to the importance of PIP2 and PA signaling pathways, other lipids such as PS, PC, PG, and CL also play critical roles in thermotolerance in plants. PS serves as a major constituent of interior surface of cell membranes involved in the maintenance of bilayer curvature and the biosynthesis of PC or PE [52,53]. PC is an abundant composition of cell membranes affecting membrane integrality and also involved in syntheses of PA, LPA, and LPC for the regulation of stress defensive responses in cells [42]. It is well known that the maintenance of higher PC content or ratio is an important strategy for plants adapting to heat stress due to its bilayer-forming characteristics [23,50]. As a main component of thylakoid membranes, PG exhibits positive role in stabilizing structure and function of chloroplasts. Significant increase in PG content conferred survival advantage in a transgenic wheat (*Triticum aestivum*) associated with less damage to thylakoid membrane and better photosynthetic capacity under heat stress [54]. Being different from the PG, the CL is mainly located in mitochondrial membranes and acts as a maintainer of functional integrity and dynamics of mitochondria in plants. Previous studies have proved that a loss-of-function *cls*
*Arabidopsis thaliana* mutant which could not synthetize the CL exhibited impaired growth and was also more sensitive to heat stress than WT [55,56]. In response to heat stress, a significant increase in endogenous Spd content improved the accumulation of PS, PC, PG, and CL in *TrSAMS*-transgenic *Arabidopsis thaliana*, which could be more beneficial to maintain membrane integrity and functionality under heat stress. In addition, *TrSAMS*-transgenic plants also showed significantly higher levels of lysophospholipids (LPA, LPI, LPC, LPE, and LPG) than WT under heat stress. Research is increasingly focusing on investigating possible functions of these lysophospholipids in plants. For example, it has been demonstrated that the LPA, LPE, and LPC regulate signal transmission, wound response, pathogen infection, or vacuolar H^+^ pool in different plants [57,58,59,60]. However, their regulatory roles in thermotolerance are still unclear and deserve to be further investigated in plant species.

Changes in lipid composition and ratio affect the formation of lipid bilayers when plants respond to high temperature. One of the important thermal adaptabilities in plant species is a possible prevention of phase transition from bilayer membranes to non-bilayer ones [20]. Heat stress causes thylakoid membrane peroxidation resulting in the disruption of chloroplast function and a subsequent decline in photosynthesis [54]. DGDG and MGDG are two abundant thylakoid lipids in plants. Although both of them play vital roles in thylakoid membrane architecture, an increase in DGDG:MGDG ratio exhibits higher contributions to the improvement of thylakoid membrane stability under abiotic stress, because DGDG is known as a bilayer-forming lipid, but the MGDG exhibits a spontaneous tendency to form non-bilayer lipid phase [61,62]. It has been proved that decreased DGDG and MGDG contents led to declines in Chl content and photochemical efficiency [63]. Higher DGDG:MGDG ratio contributed to better stability of thylakoid membrane for photosynthesis in *Arabidopsis thaliana* subjected to heat stress [64]. Enhanced heat sensitivity in *Arabidopsis thaliana* mutation with impaired biosynthesis of DGDG has also been found in a previous study [64]. *TrSAMS*-transgenic *Arabidopsis thaliana* not only had significantly higher DGDG content, but also attempted to maintain higher DGDG:MGDG ratio than WT undergoing a long period of heat stress (10 d). This could be propitious to prevent stress-induced non-bilayer phase formation, helping to retain membrane stability and integrity under heat stress.

Unlike Phls and Glls, the function of Spls in relation to plant adaptability to abiotic stress is not well documented. Most of Spls such as Cer, Hex1Cer, and CerG2GNAc1 perform as structural components of cell membranes, and some of them also act as lipid signaling molecules such as Cer for regulating cellular metabolism and renewal [21,65]. It has been elucidated that the change in Cer content affected endogenous phytohormone levels and programmed cell death (PCD) in plants [66]. Diethyl aminoethyl hexanoate (DA-6) induced the improvement in drought tolerance associated with enhanced accumulation of total Spls, Cer, Hex1Cer, and CerG2GNAc1 in leaves of white clover [67]. As compared to WT, a significant increase in endogenous Spd content was accompanied by higher Cer, Hex1, and CerG2GNAc1 levels in *TrSAMS*-transgenic *Arabidopsis thaliana* during heat stress. However, how do these Glls as structural materials affect cell membrane stability and fluidity under heat stress? What are the main downstream defensive pathways regulated by these Glls contributing to thermotolerance in plants? These questions need to be verified by further study in plant species.

## 4. Materials and Methods

### 4.1. Plant Material and Treatments

For the cultivation and treatment of white clover (cv. Ladino), seeds germinated in quartz sands and distilled water for 8 days in a growth chamber, which provided normal growth condition (23/19 °C (day/night), 14 h photoperiod, 65% relative humidity, and 700 µmol·m^−2^·s^−1^ PAR). Seedlings were then cultivated in the Hoagland’s solution [68] for 12 days and then removed carefully from quartz sands and suspended through small holes in Styrofoam boards for 7 days of hydroponic cultivation in the Hoagland’s solution. For the Spd pretreatment, 27-days-old seedlings were cultivated in the Hoagland’s solution containing Spd (70 μmol/L) for 3 days. Non-pretreated seedlings were cultivated in Hoagland’s solution without the Spd for 3 days. Spd-pretreated and untreated seedlings were then placed in normal growth chambers (mentioned above) or high temperature growth chambers (38/33 °C (day/night), 14 h photoperiod, 65% relative humidity, and 700 µmol·m^−2^·s^−1^ PAR) for 20 days. All solutions were refreshed every day. Two transgenic lines (OE4 and OE5) overexpressing a *TrSAMS* cloned from the white clover (cv. Ladino) were selected as experimental materials [69], since these two lines exhibited significantly higher endogenous Spd content than wild type under normal condition and heat stress (Figure 3B). For the cultivation and treatment of *Arabidopsis thaliana*, wild type (Col-0) and transgenic lines were grown in pots (9.5 cm length, 9.5 cm width, and 9 cm height) with peat soil and irrigated weekly with half-strength Hoagland’s solution for 20 days in a growth chamber (21/18 °C (day/ night), 70% relative humidity, 16 h photoperiod, and 350 µmol·m^−2^·s^−1^ PAR). Plants were then moved into the high temperature chambers (38/33 °C (day/ night), 70% relative humidity, 16 h photoperiod, and 350 µmol·m^−2^·s^−1^ PAR) for 10 days or kept in the normal growth chambers for 10 days. All plants were completely arranged in growth chambers. Leaf samples were collected for six independent biologic replications per treatment and twenty plants per replication were used.

### 4.2. Growth and Physiological Measurements

Chl content was detected based on the method of Barnes et al. [70]. Fresh leaves (0.15 g) were cut from plants and soaked immediately in 15 mL of dimethylsulfoxide solution at room temperature for 48 h. The absorbance of extracts was determined using a spectrophotometer (Spectronic Instruments) at 663 nm and 645 nm. EL was detected using a conductivity meter (Model 32, Yellow Springs Instrument Company). The surface of fresh leaves (0.15 g) was cleaned carefully using deionized water and then immersed in 35 mL of deionized water for 24 h. The initial conductivity (C_initial_) of soaks was detected. Leaves were then autoclaved at 120 °C for 15 min to detect the maximum conductance (C_max_) of soaks. The EL was calculated as the percentage of C_initial_/C_max_ [71]. For the determination of MDA content, fresh leaves (0.15 g) were ground with 2 mL of 50 mM cold PBS (pH 7.8) and the mixture was centrifuged at 10,000× *g* for 30 min. The supernatant (0.5 mL) was mixed with the reaction solution (1.0 mL, 20% w/v trichloroacetic acid and 0.5% w/v thiobarbituric acid) and then incubated at 95 °C for 15 min. After being centrifuged at 8000× *g* for 10 min, the absorbance of reaction solution was measured at 532 and 600 nm [72]. For PIABS and PSII Fv/Fm, a single layer of leaves was clipped for 30 min dark adaptation and readings of Fv/Fm and PIABS were recorded with the Chl fluorescence meter (Pocket PEA). SL, RL, LL, and LW were detected using 10 plants of each replicate and a total of 40 plants were used for each treatment. Fresh leaves were harvested, and FW of above-ground biomass was detected immediately. These leaves dried in an oven at 105 °C for 20 min and then at 70 °C for 3 days to get the DW of above-ground biomass.

### 4.3. Measurements of Endogenous Spermidine

Fresh leaves (0.2 g) were ground with 1.5 mL cold perchloric acid (5%, v:v) on the ice. After being incubated at 4 °C for 1 h, the homogenate was centrifuged at 10,000× *g* for 30 min (4 °C). Supernatant (0.5 mL) was mixed with 2 M NaOH (2 mL) and benzoyl chlorides (10 mL), and then the mixture was incubated at 37 °C for 30 min. Saturated NaCl solution (2 mL) and cold diethylether (2 mL) were added into the mixture in proper order and mixed by shaking. The supernatant ether phase (1 mL) was evaporated to dryness and re-dissolved in 1 mL of methanol for determination of Spd through high performance liquid chromatography (HPLC, Agilent-1200, Agilent Technologies, Santa Clara, CA, USA). Benzoyl Spd extract (20 µL) was loaded onto a reverse-phase Tigerkin^®^C18 column (150 mm × 4.6 mm, and 5 µm particle size). Mobile phase was methanol–H_2_O (64:36, v:v) and column temperature was maintained at 25 °C. The Spd peak was monitored with a UV detector at 254 nm with a flow rate of 1 mL/min [73].

### 4.4. Metabolomics Analysis

For metabolites extraction, leaf tissues (100 mg) were ground with 500 µL of prechilled solution containing 80% methanol and 0.1% formic acid. After being incubated on the ice for 5 min, the homogenate was centrifuged at 12,000× *g* for 10 min (4 °C). The supernatant was diluted by hyperpure water to final concentration containing 53% methanol and then transferred to a fresh tube for the centrifugation (15,000 *g*, 4 °C, and 10 min). Finally, the supernatant was injected into the LC-MS/MS system, which is a Vanquish UHPLC system (Thermo Fisher, Waltham, MA, USA) coupled with an Orbitrap Q Exactive series mass spectrometer (Thermo Fisher, Waltham, MA, USA). A Hyperil Gold column (C18, 100 mm × 2.1 mm, and 1.9 µm particle size) was used (a flow rate of 0.2 mL/min, 0.1% formic acid in water, and methanol). Mass spectrometers of Q Exactive series were operated in positive/negative polarity mode with spray voltage of 3.2 kV, capillary temperature of 320 °C, sheath gas flow rate of 35 arb, and aux gas flowrate of 10 arb. Raw data were processed using the Compound Discoverer 3.1 (Thermo Fisher, Waltham, MA, USA) to perform peak alignment, peak picking, and quantitation for each metabolite. Normalized data were used to predict the molecular formula based on additive ions, molecular ion peaks, and fragment ions. Peaks were matched with the mzCloud (https://www.mzcloud.org/), mzVault, and MassList database to obtain accurate qualitative and relative quantitative results. Metabolic pathways of identified metabolites were annotated using the KEGG database (http://www.genome.jp/kegg/) and Lipidmaps database (http://www.lipidmaps.org/). All metabolites with *p*-value < 0.05 and fold change ≥ 2.0 (significant increase) or ≤ 0.5 (significant decrease) were identified as DAMs.

### 4.5. Lipidomics Analysis

For lipids extraction, fresh leaves (0.15 g) were lyophilized in a Freeze Dryer for 3 days until consistent weight. Lyophilized leaves were ground into fine powders on the ice. Fine powders (20 mg) were mixed with 300 µL of chloroform and methanol (v:v, 2:1) and then shaken for 30 min. The mixture was centrifuged for 20 min at 12,000× *g* and the supernatant was removed into a new centrifuge tube. The isopropanol (300 µL) was added into the residues and then shaken for 30 min again. After being centrifuged at 12,000× *g* for 20 min, the supernatant was taken out and mixed with the previous supernatant uniformly. The mixed supernatant (200 µL) was used for the analysis of lipids by using a Vanquish UHPLC/Q Exactive Plus (Thermo Fisher, Waltham, MA, USA). Column (ACQUITY UPLC BEH C18, 100 × 2.1 mm, and 1.7 μm) temperature was maintained at 55 °C. For the mobile phase A, acetonitrile/water (60:40) with 10 mM ammonium formate and 0.1% formic acid were used. Isopropanol/acetonitrile (90:10) with 10 mM ammonium formate and 0.1% formic acid were used as the mobile phase B. Gradient or flow rate was set to 95/5~0/100 in 17 min or 0.4 mL/min, respectively. Data were acquired by software Xcalibur and identified by Lipidsearch 4.2 (Thermo Fisher, Waltham, MA, USA). Unsaturation index of total lipids, Phls, Glls, or Spls was calculated according to the formula (*n* × mol% lipid)/100 (*n* indicates the number of double bonds in the lipid molecular species and mol% indicates the composition percentage of individual lipid molecular species) [74].

### 4.6. Gene Expression Analysis

Transcript levels of genes involved in PI and PA metabolism including *AtPI4K-alpha*, *AtPIP5K1*, *AtPLC5*, *AtDGK2*, and *AtPLD-delta* were detected using real-time quantitative polymerase chain reaction (qRT-PCR). Total RNAs were extracted from fresh leaves of wild type and transgenic plants using RNeasy Mini Kit (Tiangen, Beijing, China) and then reverse-transcribed to cDNAs using a Revert Aid First Stand cDNA Synthesis Kit (Fermentas, Beijing, China). Primers were recorded in Appendix A and *AtACT2* was used as the reference gene [75,76,77,78,79]. Operation steps of qRT-PCR included 5 min at 94 °C, denaturation at 95 °C for 30 s (40 repeats), annealing at 58–61 °C for 30 s, and extension at 72 °C for 30 s (Appendix A). The formula 2^−ΔΔCT^ in the study of Livak and Schmittgen was used to calculate transcript levels of genes [80].

### 4.7. Statistical Analysis

PCA and variances of growth, physiological, and lipidomics parameters were analyzed using a statistical program (SPSS v20.0, IBM, Amunk, NY, USA). Differences among treatment means were tested using Fisher’s protected least significance (LSD) test at a 0.05 probability level. HSI was used to evaluate relative change of lipids under heat stress condition (3 h or 10 d) as compared to normal condition (0 h) based on the formula: HSI = (value of parameter under heat stress)/(value of parameter under normal condition) × 100 [81].

## 5. Conclusions

In conclusion, significant increase in endogenous Spd content through exogenous application of Spd or *TrSAMS* overexpression could effectively alleviate heat-induced growth retardation, oxidative damage to lipids, and declines in photochemical efficiency and cell membrane stability. The analysis of metabolomics showed that the main metabolic pathways were regulated by Spd in white clover including the amino acids and vitamins metabolism, biosynthesis of secondary metabolites, and lipids metabolism under high temperature condition. *TrSAMS*-transgenic *Arabidopsis thaliana* up-regulated PIP2 and PA signaling pathways and other Phls (PS, PG, PC, and CL) levels associated with better maintenance of membrane integrity and functionality in response to heat stress. In addition, *TrSAMS*-transgenic *Arabidopsis thaliana* also could maintain higher DGDG level and DGDG:MGDG ratio, which was beneficial to retain bilayer phase contributing to thylakoid membrane stability and integrity undergoing a long period of heat stress.

## Figures and Tables

**Figure 1 ijms-23-12247-f001:**
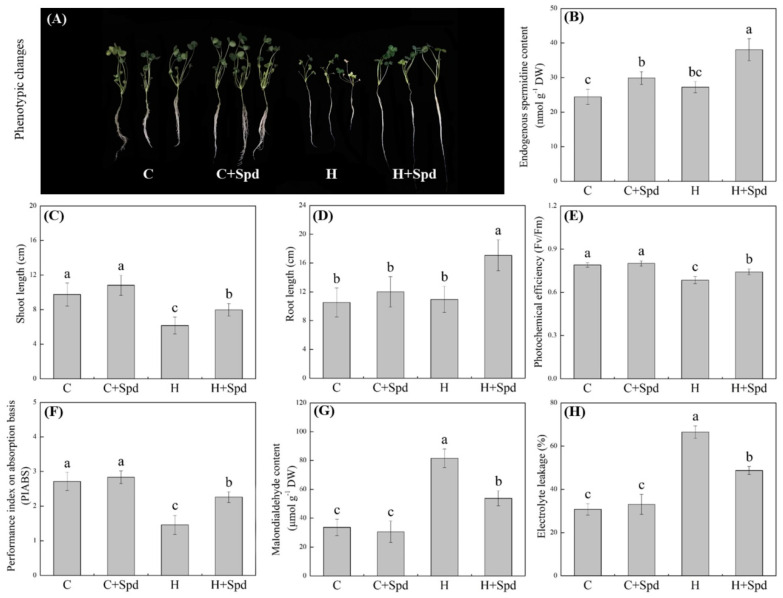
Changes in (**A**) phenotype, (**B**) endogenous spermidine (Spd) content, (**C**) shoot length, (**D**) root length, (**E**) photochemical efficiency, (**F**) performance index on absorption basis, (**G**) malondialdehyde content, and (**H**) electrolyte leakage in leaf of white clover in response to exogenous Spd and heat stress. Vertical bars indicate ±SE of mean (*n* = 4) and different letters above column indicate significant differences (*p* ≤ 0.05) among treatments.

**Figure 2 ijms-23-12247-f002:**
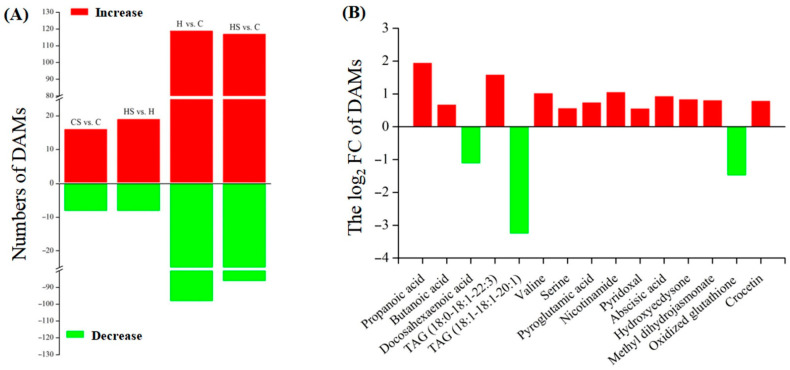
Changes in (**A**) the total numbers of differentially accumulated metabolites (DAMs) in different groups (CS vs. C, HS vs. H, H vs. C, and HS vs. C) and (**B**) key DAMs in the HS vs H. C, control; CS, control + spermidine; H, heat stress; HS, heat stress + spermidine.

**Figure 3 ijms-23-12247-f003:**
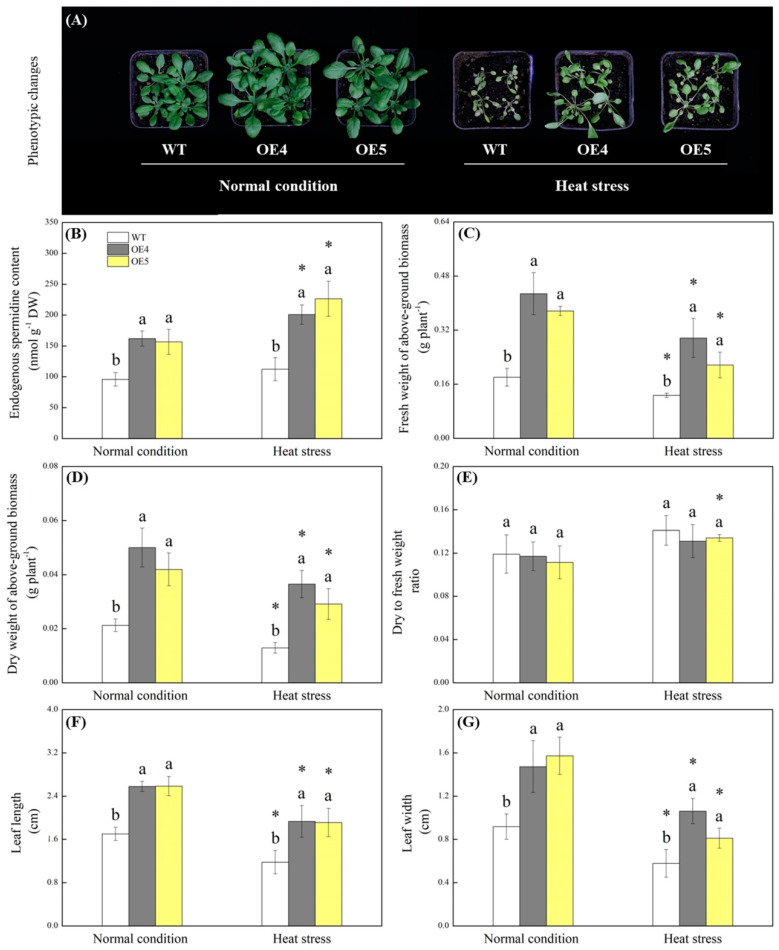
Changes in (**A**) phenotype, (**B**) endogenous spermidine content, (**C**) fresh weight of above-ground biomass, (**D**) dry weight of above-ground biomass, (**E**) dry to fresh weight ratio, (**F**) leaf length, and (**G**) leaf width of wild type (WT) and transgenic *Arabidopsis thaliana* overexpressing the *TrSAMS* (OE4 and OE5) involved in spermidine biosynthesis under normal condition and heat stress. Vertical bars indicate ±SE of mean (*n* = 4). Different letters above column indicate significant differences (*p* ≤ 0.05) among treatments under a particular condition (normal condition or heat stress). The asterisk “*” indicates significant differences (*p* ≤ 0.05) for a particular material (WT, OE4, or OE5) for comparison between normal condition and heat stress.

**Figure 4 ijms-23-12247-f004:**
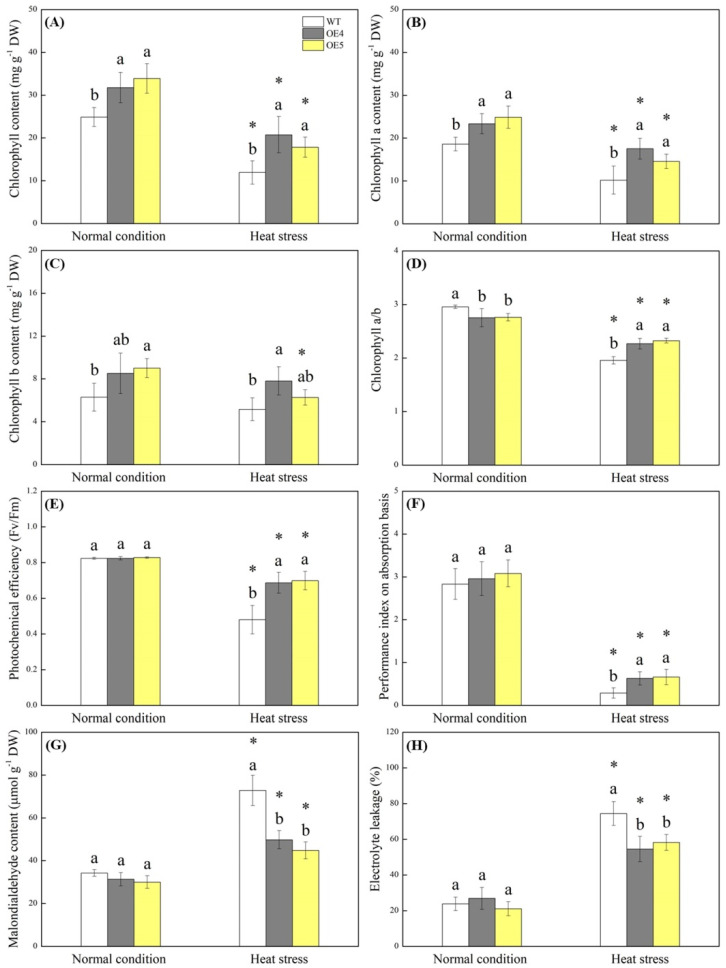
Changes in (**A**) total chlorophyll content, (**B**) chlorophyll a content, (**C**) chlorophyll b content, (**D**) chlorophyll a/b ratio, (**E**) photochemical efficiency, (**F**) performance index on absorption basis, (**G**) malondialdehyde content, and (**H**) electrolyte leakage of wild type (WT) and transgenic *Arabidopsis thaliana* overexpressing the *TrSAMS* (OE4 and OE5) involved in spermidine biosynthesis under normal condition and heat stress. Vertical bars indicate ±SE of mean (*n* = 4). Different letters above column indicate significant differences (*p* ≤ 0.05) among treatments under a particular condition (normal condition or heat stress). The asterisk “*” indicates significant differences (*p* ≤ 0.05) for a particular material (WT, OE4, or OE5) for comparison between normal condition and heat stress.

**Figure 5 ijms-23-12247-f005:**
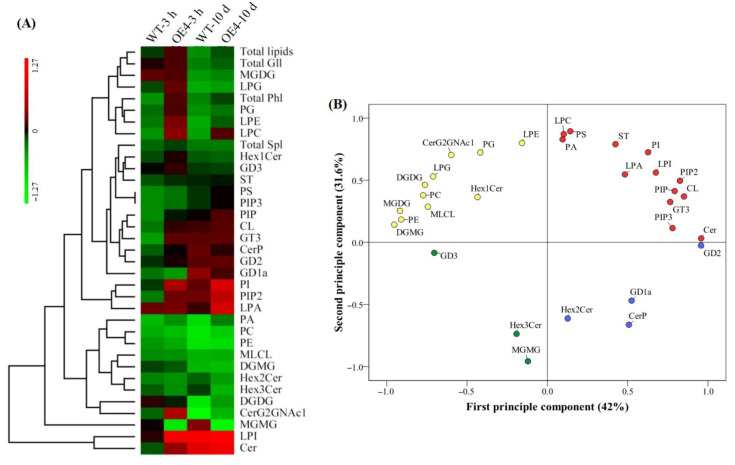
Changes in (**A**) heatmap as demonstrated by log_2_ (fold change in contents of total lipids, lipid groups, and lipid classes) and (**B**) principal component analysis (PCA) based on 31 different lipid classes in leaves of wild type (WT) and transgenic *Arabidopsis thaliana* (OE4) overexpressing the *TrSAMS* involved in spermidine biosynthesis in response to spermidine and heat stress. The heat map was made based on log_2_ fold change of heat-stressed plants (3 h and 10 d) as compared to non-stressed plants (0 h).

**Figure 6 ijms-23-12247-f006:**
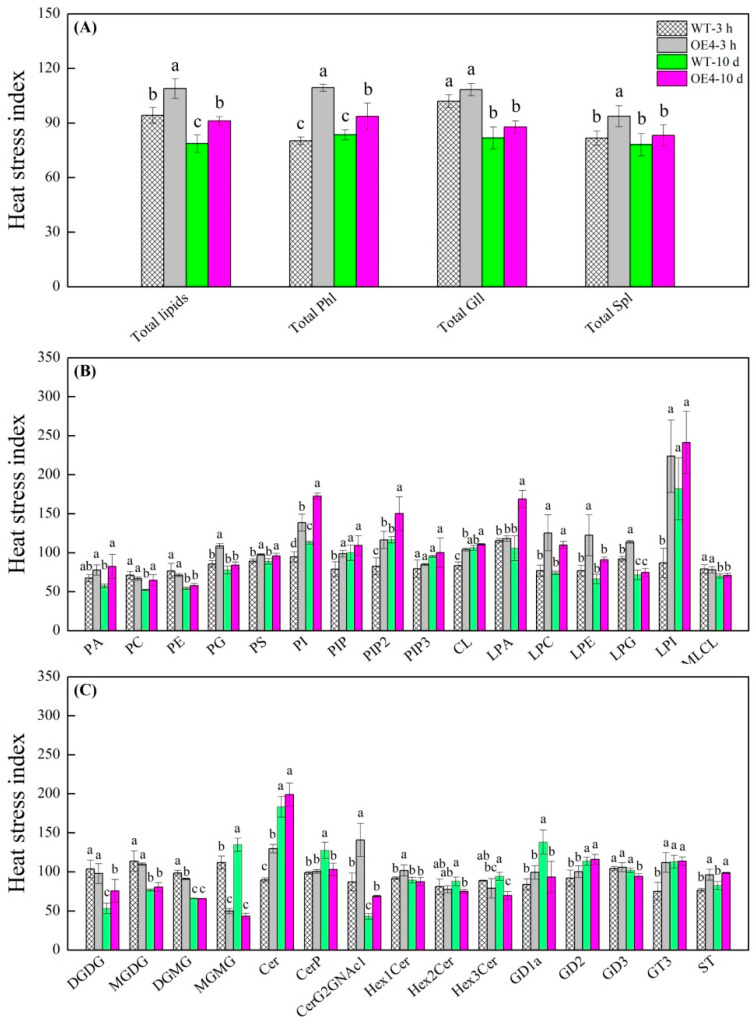
Changes in heat stress index of (**A**) total lipids, total phospholipids (Phl), total glycoglycerolipids (Gll), and total sphingolipids (Spl), (**B**) different Phl classes, and (**C**) different Gll and Spl classes in leaves of wild type (WT) and transgenic *Arabidopsis thaliana* (OE4) overexpressing the *TrSAMS* involved in spermidine biosynthesis during heat stress. Vertical bars indicate ±SE of mean (*n* = 4) and different letters above column indicate significant differences (*p* ≤ 0.05) among treatments.

**Figure 7 ijms-23-12247-f007:**
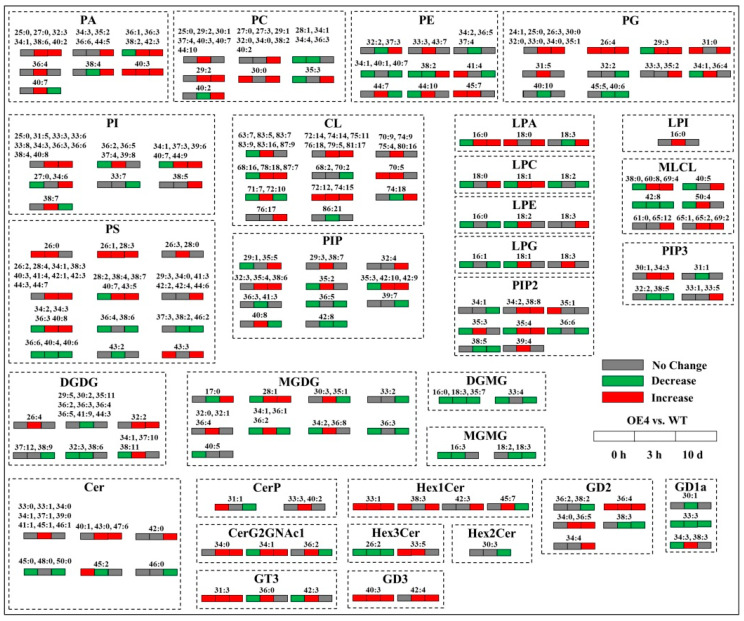
Changes in lipid molecular species of different lipid classes in leaves when the transgenic *Arabidopsis thaliana* overexpressing the *TrSAMS* was compared with the wild-type (WT) *Arabidopsis thaliana* during heat stress (OE4 vs. WT). Red indicates a significant up-regulation, green indicates a significant down-regulation, and gray indicates no significant change.

**Figure 8 ijms-23-12247-f008:**
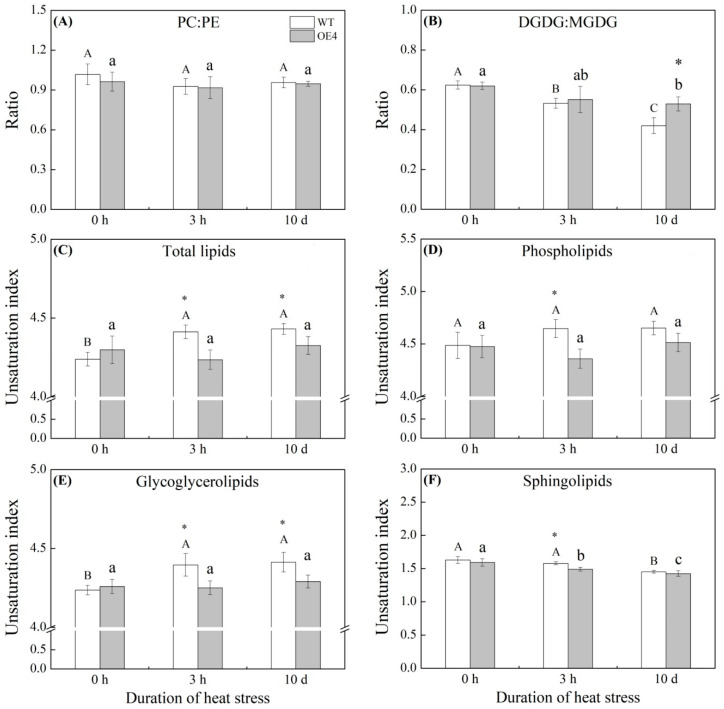
Changes in (**A**) the ratio of PC:PE, (**B**) the ratio of DGDG:MGDG, (**C**) the unsaturation index of total lipids, (**D**) the unsaturation index of phospholipids, (**E**) the unsaturation index of glycoglycerolipids, and (**F**) the unsaturation index of sphingolipids in leaves of wild type (WT) and transgenic *Arabidopsis thaliana* overexpressing the *TrSAMS* involved in spermidine biosynthesis during heat stress. Different letters above column indicate significant differences (*p* ≤ 0.05) in a particular material (WT or OE4) from 0 h to 10 d. The asterisk “*” indicates significant differences (*p* ≤ 0.05) between WT and OE4 at one particular time point (0 h, 3 h, or 10 d).

**Figure 9 ijms-23-12247-f009:**
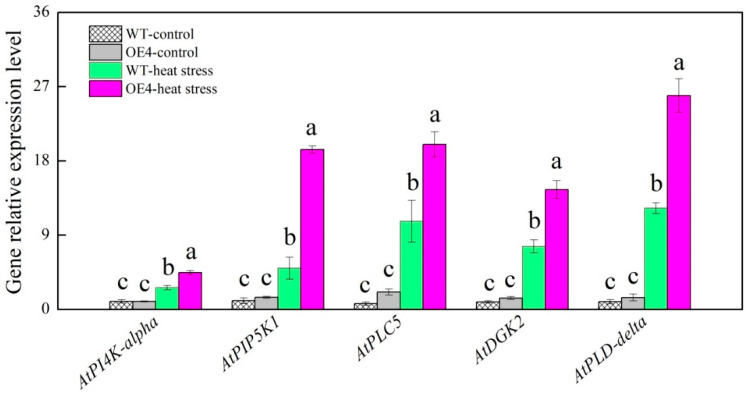
Changes in key genes involved in metabolic pathways of phosphoinositide (PI) and phosphatidic acid (PA) regulated by spermidine in leaves of *Arabidopsis thaliana* during heat stress. Vertical bars indicate ±SE of mean (*n* = 4) and different letters above column indicate significant differences (*p* ≤ 0.05) among treatments.

**Figure 10 ijms-23-12247-f010:**
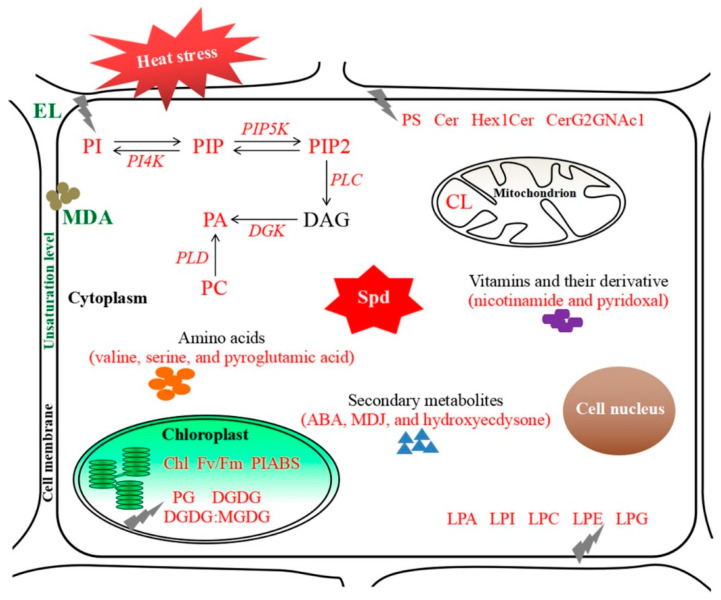
Regulatory pathways induced by the spermidine associated with alterations in metabolic homeostasis, lipids signaling, and lipidomic remodeling in plants under heat stress. Red or green characters indicate significant increases or decreases induced by spermidine under heat stress, respectively.

## Data Availability

Not applicable.

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
