# Peer review of "Metabolic Regulation and Lipidomic Remodeling in Relation to Spermidine-induced Stress Tolerance to High Temperature in Plants"

_ijms, 2022, doi:10.3390/ijms232012247_

Round 1

Reviewer 1 Report

In the present study “Metabolic regulation and lipidomic remodeling in relation to spermidine-induced stress tolerance to high temperature in plants”, authors provide a novel insight into the function of Spd against heat stress through regulating lipids signaling and reprograming in plants. I think that the work falls into the scope of the journal and findings are interesting, however MS demands minor revision.

Comments:

Abstract: This section seems scary and unclear. I would suggest to add materials and methods section in the abstract.

Introduction: The significant novel point of the study over the precedent studies is not clear. There are two major concerns with this MS. First one is grammatical mistakes, language error, typographical mistakes. Aims and objectives of the study are not clear. Less references have been reviewed, more related references can be cited e.g.  PLoS ONE, 2015, 10(4): e0123328; Ecotoxicology and Environmental Safety, 2014, 110: 197–207; and Biologia Plantarum, 2014, 58(1): 131-138.

Materials and methods: Avoid to use “we, our, us”. How many replications per treatment? How many plants per replication? Drafting of many sentences need to be revised. Plz standardize hr for hour/hours. Please thoroughly standardize L for liter. 

Results and Discussion: In results, there is a striking lack of connectors between sentences and leading to confusing. Many sentences are useless in Results section, please focus your key findings. In first use, mention the full names, then you can use abbreviations. I would suggest to present your results by increase/decrease %age. One way of improving Discussion is to avoid repetition of results in this part. Discussion is very shallow and need in depth discussion with the recent literature published. In discussion, there is a lack of mechanistic approach.

Author Response

In the present study “Metabolic regulation and lipidomic remodeling in relation to spermidine-induced stress tolerance to high temperature in plants”, authors provide a novel insight into the function of Spd against heat stress through regulating lipids signaling and reprograming in plants. I think that the work falls into the scope of the journal and findings are interesting, however MS demands minor revision.

Response: Thank you very much for your professional and careful review. We have revised the manuscript according to suggestions.

Comments:

Abstract: This section seems scary and unclear. I would suggest to add materials and methods section in the abstract.

Response: Thank you for your suggestion. We have added materials and methods section in the abstract (line 13-20).

Introduction: The significant novel point of the study over the precedent studies is not clear. There are two major concerns with this MS. First one is grammatical mistakes, language error, typographical mistakes. Aims and objectives of the study are not clear. Less references have been reviewed, more related references can be cited e.g.  PLoS ONE, 2015, 10(4): e0123328; Ecotoxicology and Environmental Safety, 2014, 110: 197–207; and Biologia Plantarum, 2014, 58(1): 131-138.

Response: Significant novel point of the study over the precedent studies has been discussed in the Introduction section (line 83-85). Aims and objectives of the study have been demonstrated in the last paragraph of the Introduction (line 86-95). The manuscript has been improved according to suggestions and these references which reviewer mentioned above also have been added in revised manuscript (line 62).

Materials and methods: Avoid to use “we, our, us”. How many replications per treatment? How many plants per replication? Drafting of many sentences need to be revised. Plz standardize hr for hour/hours. Please thoroughly standardize L for liter.

Response: We have deleted the “we, our, us” in this section. Six replications per treatment were used. Twenty plants per replication were used in this study. These information have been added in the section of Materials and methods. The “hr” has been changed to “hour/hours”.

Results and Discussion: In results, there is a striking lack of connectors between sentences and leading to confusing. Many sentences are useless in Results section, please focus your key findings. In first use, mention the full names, then you can use abbreviations. I would suggest to present your results by increase/decrease %age. One way of improving Discussion is to avoid repetition of results in this part. Discussion is very shallow and need in depth discussion with the recent literature published. In discussion, there is a lack of mechanistic approach.

Response: The manuscript has been improved. We have checked and added full names, then used abbreviations throughout the manuscript. Yes, we only briefly described key findings in the results section. Significant changes in parameters have been described by using increase/decrease % in the results section. Some important results should be mentioned in the Discussion section in order to let readers understand what we found in current study. Discussion section has clearly demonstrated the key mechanism regulated by the Spd in plants, as summarizing in the Fig. 10.

Reviewer 2 Report

General Comments

Reviewed is the manuscript “Metabolic regulation and lipidomic remodeling in relation to spermidine-induced stress tolerance to high temperature in plants” submitted by Zhou Li, et, al. The authors used the metabolomics/lipidomics approach to evaluate the relation to spermidine-induced stress tolerance to high temperature in plants and concluded that the main metabolic pathways regulated by the Spd in cool-season white clover under heat stress. This study has been performed using the state-of-the-art methodology and bioinformatics analysis, very clear and well written. These results obtained at a systemic level are impressive and need now to be validated at a higher scale. Overall, the authors clearly demonstrate their approach and detail the performance gained in this research field and the article meets the required standards for publication after minor edits.

Specific Comments

-           Given the large number of tests performed, the multiple testing correction (e.g., FDR) becomes necessary. It is important that this is addressed.

-           The authors mentioned that “Metabolites were annotated by using the KEGG database (http://www.genome.jp/kegg/) and Lipidmaps database (http://www.lipid-maps.org/).” Are all the annotations putative? Since the authors mentioned they have used LC-MS/MS system, tandem MS data is recommended.

-           Figure 1, is there any significant difference among those groups? If so, adding a significant difference level is recommended.

- In figure 5, consider adding more details to the figure legend. Panel A, data used to plot the heatmap should be stated clearly (lipid groups?).

Author Response

General Comments

Reviewed is the manuscript “Metabolic regulation and lipidomic remodeling in relation to spermidine-induced stress tolerance to high temperature in plants” submitted by Zhou Li, et, al. The authors used the metabolomics/lipidomics approach to evaluate the relation to spermidine-induced stress tolerance to high temperature in plants and concluded that the main metabolic pathways regulated by the Spd in cool-season white clover under heat stress. This study has been performed using the state-of-the-art methodology and bioinformatics analysis, very clear and well written. These results obtained at a systemic level are impressive and need now to be validated at a higher scale. Overall, the authors clearly demonstrate their approach and detail the performance gained in this research field and the article meets the required standards for publication after minor edits.

Response: Thank you very much for your professional and careful review. We have revised the manuscript according to suggestions.

Specific Comments

- Given the large number of tests performed, the multiple testing correction (e.g., FDR) becomes necessary. It is important that this is addressed.

Response: Thanks. The FDR is often used to analyze a large number of data such as transcriptome for screening differentially expressed genes. In our current study, lipidomics only includes 31 lipids. The “Fisher’s protected least significance (LSD) test at a 0.05 probability level” was used to analyze these data in our current study.

- The authors mentioned that “Metabolites were annotated by using the KEGG database (http://www.genome.jp/kegg/) and Lipidmaps database (http://www.lipid-maps.org/).” Are all the annotations putative? Since the authors mentioned they have used LC-MS/MS system, tandem MS data is recommended.

Response: Thank you for your careful review. Yes, we used LC-MS/MS system and tandem MS data to separate and identify metabolites. The KEGG database (http://www.genome.jp/kegg/) and Lipidmaps database (http://www.lipid-maps.org/) were used to analyze metabolic pathways of identified metabolites. The sentence “Metabolites were annotated by ...” Has been changed to “Metabolic pathways of identified metabolites…”.

- Figure 1, is there any significant difference among those groups? If so, adding a significant difference level is recommended.

Response: Thanks. A significant difference level has been added in revised manuscript.

- In figure 5, consider adding more details to the figure legend. Panel A, data used to plot the heatmap should be stated clearly (lipid groups?).

Response: Thanks. Necessary information in details has been added in the legend of figure 5 according to suggestions.